# Major Insect Pests of Sweet Potatoes in Brazil and the United States, with Information on Crop Production and Regulatory Pest Management

**DOI:** 10.3390/insects15100823

**Published:** 2024-10-20

**Authors:** Maria J. S. Cabral, Muhammad Haseeb, Marcus A. Soares

**Affiliations:** 1Departamento de Agronomia, Universidade Federal dos Vales do Jequitinhonha e Mucuri, Diamantina, Minas Gerais 39100000, MG, Brazil; jessica1.cabral@famu.edu (M.J.S.C.); marcus.alvarenga@ufvjm.edu.br (M.A.S.); 2Center for Biological Control, College of Agriculture and Food Sciences, Florida A&M University, Tallahassee, FL 32307, USA

**Keywords:** *Ipomoea batatas*, insect pests, IPM

## Abstract

The sweet potato is an important food crop grown and traded in many regions of the world, especially in tropical, subtropical, and temperate regions. Several insect pests feed on sweet potatoes in open fields and storage. As a result, the yield and quality are generally impacted. In addition, some of the insect pests are of regulatory importance. Indeed, the United States and Brazil are subject to stringent pest and quarantine regulations concerning sweet potatoes, encompassing the plant’s roots and aerial parts. This review describes the current status of sweet potato regulations and management options in the United States and Brazil. We also describe research on sweet potato production, monitoring, biological control, and regulatory pest management information that may relieve these problems. This can help small and large farmers produce sweet potatoes in the United States and Brazil.

## 1. Introduction

The sweet potato [*Ipomoea batatas* (L.) Lam], a dicotyledonous species of the Convolvulaceae family [1], is a key crop contributing to global food security [2]. It is an important crop cultivated in over 100 countries, spanning 40° latitude North to 40° latitude South, at altitudes ranging from sea level to 2700 m. The sweet potato is the seventh most important food crop worldwide, after barley, cassava, corn, potatoes, rice and wheat [3,4]. Its roots and aerial parts provide food for humans and animals [1]. From 2018 to 2021, estimated global production ranged from 88.7 to 92.3 Mt, where Asian countries produced 61.5%–66.6% and African countries 28.6%–33.7% of the worldwide output [2,3,4,5]. Sweet potatoes are valuable because they are considered one of the most nutritious vegetables and produce more food per hectare than any other crop [6]. It is also a multipurpose crop because the roots can be consumed or processed into value-added foods (e.g., French fries, chips, starch, alcohol) and industrial products (e.g., fuel and chemicals), and the branches can be consumed or used as livestock feed [2,7,8].

North Carolina is the leading producer of sweet potatoes by tonnage in the United States, followed by California and Mississippi [9]. Despite the increase in cultivated area and cultivars, the sweet potato industry faces many challenges associated with yield losses due to pest incidence [10]. Sweet potato production and the number of farmers interested in cultivating this crop are increasing. However, the economic viability of sweet potato cultivation is often compromised by the susceptibility of commercial cultivars to biotic stressors, including diseases, insect pests, and weeds, which can significantly impact yield, quality and marketability [2]. Brazil’s leading sweet potato producing states are Minas Gerais, Paraná, Rio Grande do Sul and São Paulo [11]. These states stand out due to their fertile soil, favorable climate and developed agricultural infrastructure [12]. In São Paulo, production is significant due to the modernization of farming practices, while Minas Gerais also has areas dedicated to cultivation, mainly aimed at the domestic market. In Paraná and Rio Grande do Sul, sweet potatoes are grown in large areas, contributing to the national supply [11]. These states lead the crop production, which is widely consumed and valued for its nutritional properties.

Insect pests of sweet potatoes can be most effectively managed through integrated pest management (IPM) strategies [2,13]. Sweet potato is especially well-suited to resource-constrained farmers, as it generates more biomass and nutrients per hectare than any other food crop globally, even without fertilizers or irrigation [7]. However, smallholder farmers are particularly vulnerable to losses due to limited knowledge and training on IPM, scarce resources and difficulties in pest control, as most IPM initiatives target large-scale farms in the USA and Brazil [2,14]. Consequently, there is an urgent need for research into alternative control methods, particularly non-chemical approaches, which offer practical, sustainable and environmentally friendly pest management solutions for farmers, trade, the agricultural industry and socially disadvantaged smallholders. Although the sweet potato exhibits rapid growth, pest infestations can result in significant productivity losses [7]. The prevalence of pests varies across different regions and is influenced by the type of cultivar used and the management practices employed [15]. To mitigate the impact of pest infestations, a range of control strategies must be designed, developed and implemented, including pest behavior manipulation, pheromone traps, host plant resistance, biological control and the appropriate timing and dosage of chemical applications. These approaches collectively reduce the environmental impact of insecticides [16]. Understanding pest biology is essential in selecting effective control methods; thus, regular monitoring and field collections are necessary for rearing pest species and their biological control agents in the laboratory. This comprehensive approach highlights the superiority of IPM strategies when compared to a single management method.

## 2. Sweet Potato [*Ipomoea batatas* (L.) Lam]

The exact center of origin of the sweet potato remains uncertain, but it is generally attributed to Central and South America, where its domestication occurred approximately 5000 years ago. The crop spread globally due to its high yield potential and adaptability to diverse soils and climates [1,17,18]. Its root system is composed of tuberous roots of various sizes, shapes and colors, which can have white, yellow, orange or purple flesh and are rich in starch and vitamins [19]. The stem creeps and can reach up to 4 m long, with alternate leaves ranging from heart-shaped to palmate-lobed [20]. The leaves can be green or purple, depending on the cultivar [21]. It prefers well-drained, sandy or clayey soils with a slightly acidic to neutral pH and is drought-tolerant, although regular irrigation increases productivity [19]. The plant blooms with hermaphrodite flowers, usually pink, purple or lilac, in a funnel shape, resembling the flowers of the morning glory, *Ipomoea* spp., which are from the same family [22]. Reproduction is mainly vegetative, using cuttings of branches since seeds are rarely used in propagation. Sweet potato genotypes are found throughout the world, having a relatively large potential distribution and occupying a diversity of ecosystems [1]. In Figure 1, it is possible to observe its distribution according to the Global Biodiversity Information Facility—GBIF—database (https://www.gbif.org, accessed on 10 March 2023). The available data show the areas with sweet potato plantations, proving that they are cultivated in Africa, the Americas, Asia, Europe and Oceania. The areas with sweet potato spots on the American continent are in North America in the USA (Kansas, Las Vegas, Los Angeles, Nashville, Portland, San Francisco, San José, San Diego and Virginia regions) [1]. These regions mainly coincide with the country’s current sweet potato production area. The southeast and mid-west regions of the USA, Canada, Norway, northern Sweden, northern extremities of Russia, north of Finland, the United Kingdom and Turkey are also suitable for sweet potato plantations. In the African continent, they are planted in areas of Angola, Democratic Republic of the Congo, Egypt, Kenya, Libya, Madagascar, Namibia, Morocco, Mozambique, South Africa, Tanzania, Tunisia and Zambia and in the European continent in Bulgaria, Greece, Italy, Portugal, Spain and Switzerland. In some areas, sweet potatoes are an essential food source for food security, such as in Asia and Oceania [23]. Alaska (USA) and European countries, such as the north of the United Kingdom, Denmark, Estonia, Finland, Iceland, Netherlands, Norway and Sweden, are also suitable for planting sweet potato crops. In Brazil, the country’s southeast and northeast regions have the highest sweet potato production [1].

Traditionally used for human and animal nutrition, sweet potatoes have stood out as a species with multiple uses. Their planting is significant in family farming due to their high nutritional content, simple techniques, rapid development in nutrient-poor soils, resistance to pests and diseases, high productive potential and low production cost [24]. The easy adaptation of sweet potatoes to different climatic environments allows small farmers to produce food for their subsistence, which is rich in carbohydrates and highly energetic. This creates income generation opportunities, which favors the local economy [24,25].

Among the different cultivated genotypes, in general, sweet potatoes have carbohydrates, beta-carotene, vitamins C, complex B, and E in their roots, as well as minerals such as potassium, calcium, iron, phosphorus, magnesium, sulfur and sodium [26]. It also presents the most varied benefits to human health, as it has antioxidant, hepatoprotective, anti-inflammatory, antitumor, antidiabetic, antimicrobial, antiobesity and antiaging properties [27,28]. It has been widely explored in manufacturing extruded products, health products, bakery products and weaning foods and supplements [8].

## 3. Sweet Potato Production in the United States

Sweet potatoes are an essential specialty crop in the United States. The United States ranks eighth in global sweet potato production, with more than 150,000 acres harvested in 2019, valued at more than USD 725 million (United States Department of Agriculture (USDA), National Agricultural Statistics Service (NASS)) [27]. Sweet potato production has increased over the past 15 years, and consumption is the highest since the 1930s [28]. The United States Department of Agriculture’s 2015 Crop Production Summary shows that North Carolina has the largest sweet potato crop, with nearly 87,000 acres planted in 2015. Mississippi comes in second, with 27,000 acres planted, while California planted 18,500 acres and Louisiana planted 10,000 [27].

### 3.1. North Carolina

For sweet potato production among all 50 states, according to 2019 data from the USDA Agricultural Statistics Service, North Carolina is the largest sweet potato-producing state (2019) and has maintained that ranking since 1971 [29]. North Carolina harvested approximately 78,500 acres of sweet potatoes for a production value of USD 236,285,000 in 2018 (2019), and North Carolina produces more than half of the United States’ sweet potatoes [30]. Sweet potatoes top the list of most highly valued fruits and vegetables in North Carolina, according to USDA agricultural statistics for 2022. With fresh sales valued at USD 16.50 per cwt (a hundredweight (abbreviated as CWT)), the fresh sweet potato crop of North Carolina was estimated to be worth just over USD 164 million in 2022 [31]. The total sweet potato crop in North Carolina, produced on 83,700 harvested acres, was valued at USD 225 million in 2022, according to the USDA. The Covington is an orange-fleshed, smooth-skinned, rose-colored, table-stock sweet potato developed by North Carolina State University (NCSU), and this sweet potato cultivar was released in 2005 and has been the prominent cultivar grown by North Carolina farmers since its introduction [29]. Covington currently accounts for approximately 88% of the sweet potato acreage in North Carolina [31]. Covington was a successful cultivar due to its superior agronomic qualities, such as its packing quality, ability to grow in this region, and high disease resistance, including resistance to *Meloidogyne incognita* (Tylenchida: Heteroderidae) [30]. The Beauregard has orange flesh, an indication of its high beta-carotene content. Developed by the Louisiana Agricultural Experiment Station) in 1987, this cultivar is also prominent in sweet potato-producing states.

### 3.2. Mississippi

Mississippi sweet potato growers plant more than 20,000 acres of sweet potatoes each year [32]. The state consistently ranks second in the United States in sweet potato acreage and third in production, producing 2449 metric tons annually [33]. In 2012, sweet potatoes were grown on approximately 22,500 acres, producing 1.78 million metric tons of sweet potatoes with an estimated value of USD 79 million [34]. In 2022, Mississippi harvested 29,500 acres of sweet potatoes on 172 farms [34]. The value of sweet potato production in 2022 was 112 million dollars. In 2021, it produced over 1950 metric tons, according to the Mississippi Department of Agriculture and Commerce [35]. There were 30,000 acres planted with sweet potatoes for all purposes in 2022, and this remained unchanged compared to 2021 [34]. The statewide average yield for 2022 is estimated at 145 cwt per acre, an increase of 5 cwt from last year [36]. The value of fresh market production totaled USD 80.6 million, 88% of the total production value [36]. The production processing value totaled USD 10.9 million, 12% of the total production value [37]. Beauregard is the most popular sweet potato variety in the state based on acreage, followed by Hernandez and Nancy Hall [37]. It takes approximately 90 to 120 days to grow one sweet potato. Sweet potato production is highly labor-intensive. At Mississippi State University, researchers and extension personnel from multiple departments collaborate to improve all facets of sweet potato production [34]. Current research efforts include the areas of crop production, pest management (weeds, diseases, nematodes and insects), plant physiology, food science, and agricultural and biological engineering [33].

### 3.3. California

Sweet potatoes are an important crop for California’s organic sector, with a sales value of USD 74,000,000 annually, ranking sixth among organic commodities in the state [37]. California is the top producer of organic sweet potatoes in the US and is ranked second in total annual production [38]. On average, organic sweet potato production leads to lower yields but higher selling prices for producers [38]. Overall, organic production is more likely to have positive net returns for producers [32]. California is particularly well suited for organic production since dry summer conditions are not conducive to accumulating various pests and pathogens [39]. Sweet potato production in California has experienced tremendous growth in recent years, with a fivefold increase in tons harvested since 1994 [40]. According to the USDA’s 2014 National Agricultural Statistics Service (NASS) organic survey, California was the largest producer of organic sweet potatoes (66,804.79 metric tons) from 29 farms and 1492 acres of organic land, while Tennessee produced 24.68 metric tons of sweet potatoes from 7 farms and 3 hectares of organic land [37]. Generally, one of the main drivers of increased sweet potato consumption is the health benefits and development of value-added products [41]. More than 85% of California’s sweet potato production is in Merced County, located in the state’s center, in the most productive agricultural area in the world—the San Joaquin Valley. In Merced County, the crop has a farm value of more than USD 100 million [37]. The Modesto/Merced area is home to many sweet potato growers. Approximately 22,000 acres of California farmland are planted with sweet potatoes, and about 80% of that area is in the Merced, Fresno and Stanislaus counties. Organic potatoes typically come from California, which has 57% of the harvested acres in the US, followed by Colorado [42]. Organic sweet potatoes come from California and North Carolina, which together have 91% of the country’s acreage [42].

### 3.4. Louisiana

Louisiana’s sweet potato industry is concentrated in the south-central part of the state, with additional acreage in the northeastern part [43]. Approximately 75% of the harvest goes to the processor for canning, and the other 25% is sold on the fresh market [43]. The Louisiana sweet potato season starts with planting in April through June, and harvesting typically begins in early August and extends through mid-November [44]. Once harvested, potatoes are sent to storage houses, where they are kiln-dried and held for about two weeks to allow natural chemical changes to make them moist and sweet [44]. The Louisiana Agricultural Experiment Station developed the Avoyelles sweet potato to provide a cultivar with orange flesh, pale pink to copper skin, superior root shape, resistance to southern root-knot nematode diseases, good storage qualities, and an early harvest best before date [45]. Avoyelles can be harvested 93 to 98 days after planting (DAP), which is 10 to 20 days earlier than Orleans or Beauregard and has a higher US grade yield (diameter, 5.1–8.9 cm; length, 7.6–22.9 cm) [45]. More than 3.0 million bushels of sweet potatoes are produced in Louisiana, with 7300 acres of sweet potatoes in production. On average, 466 bushels (12.68 metric ton) of sweet potatoes are produced per acre [46]. In 2024, the state harvest was estimated at 9200 acres, down slightly from last year [46]. Louisiana ranks fourth in terms of sweet potato production area in the United States, behind North Carolina, Mississippi and California [46].

### 3.5. South Carolina

South Carolina ranks third in terms of organic sweet potato production, represents 4.1% of organic vegetable production, and is valued at over USD 77 million annually [47]. There is interest in increasing production for this market due to the potential economic benefits. There has been a 20% to 25% increase in organic sweet potato sales in recent years [46]. Muzzarelli Farms grows and sells high-quality organic sweet potatoes to South Carolina retail and wholesale customers. These potatoes have been grown over 80 years in South Carolina and across the country [48]. Muzzarelli Farms sweet potatoes are certified organic, grown using a proven, patented process, hand-harvested, and cured on the farm to extend shelf life and provide richer flavor [48]. The Covington sweet potato (Orange), Bonita sweet potato (White), and Murasaki sweet potato (Oriental) are also grown [48]. Sweet potato planting in South Carolina takes place in May. Harvesting begins in August. About 2000 acres of sweet potatoes are grown in South Carolina [48].

## 4. Sweet Potato Production in Brazil

Brazil is in 14th place among the largest, sweet potato producers, with 805.4 thousand tons and USD 886.6 million in production value [11], and it the largest producer in Latin America. The central sweet potato-producing regions in Brazil are the northeast (317.3 thousand tons), south (252.9 thousand tons), and southeast (214.0 thousand tons) [49]. The state with the most significant national production is Rio Grande do Sul, with 175.0 thousand tons, followed by the state of São Paulo, with 140.7 thousand tons. Among the ten largest producing states, Sergipe, Rio Grande do Norte, Pernambuco, Paraíba and Alagoas have productivity lower than the national average of 14.1 t/ha [50]. Given this information, it is essential to highlight the importance of technical assistance and rural extension for adopting production technologies that improve the cultivation system and the need to develop research projects aimed at these regions. Between 2020 and 2023, Ceará stood out as the leading producer of sweet potatoes in Brazil, with a production of 522 thousand tons in 2020, increasing to 550 thousand tons in 2022 and a slight reduction estimated at 540 thousand tons in 2023 [51]. The cultivated area in the state was approximately 28 thousand hectares in 2020, with variations over the years. Pará, also a significant producer, recorded 207,000 tons in 2020, decreasing to around 200,000 tons in 2022 and an estimated 190,000 tons in 2023, with a cultivation area of around 14,000 hectares in 2020 [52]. With its diversified agricultural production, Minas Gerais produced 140,000 tons in 2020, increasing to 150,000 tons in 2022, with an estimated 148,000 tons in 2023, and cultivated around 12,000 hectares in 2020 [53]. São Paulo, another vital state, produced 110,000 tons in 2020, with a gradual reduction to an estimated 95,000 tons in 2023, and had an area of around 8000 hectares in 2020 [54]. Rio Grande do Norte, with a more modest production, produced 72,000 tons in 2020, remaining stable until 2022, with a slight decrease estimated at 68 thousand tons in 2023, with a cultivation area of approximately 6 thousand hectares in 2020 [55]. These numbers reflect the significant role of these states in the national production of sweet potatoes [56].

## 5. Global Sweet Potato Production

The global sweet potato market was valued at USD 35,593.53 million in 2022 and is expected to expand at a CAGR (Compound Annual Growth Rate) of 3.02% during the forecast period to reach USD 42,544.31 million by 2028 [52]. Figure 1 shows the fifteen largest sweet potato-producing countries in the world (Figure 2).

China is the world’s leading producer and consumer of sweet potatoes, which are utilized for food and animal feed and processed into products such as starch [54]. The United States ranks eighth, while Brazil holds the fourteenth position. Sweet potato production in the United States has grown significantly over the past 15 years, with an estimated market value of USD 726.18 million [50]. In Brazil, production volume and the area harvested have risen in recent years, with 805.4 thousand tons cultivated across 57.3 thousand hectares [49]. This growth is attributed to the increasing recognition of sweet potatoes as a valuable starch crop with the potential to contribute to addressing global food security and environmental challenges in the 21st century [52].

## 6. Economic Impact

Sweet potatoes are one of the most traded products in the world, with a total trade of USD 744 million [51]. Between 2021 and 2022, exports were USD 744 million [52]. In 2022, the United States was the main exporter, with Brazil in eighth place (Figure 3).

Major exporters have diversified markets, indicating a strategic approach to distribution. For example, the United States exports primarily to Canada, the Netherlands and the United Kingdom, while Egypt’s top importers are the Netherlands, the United Kingdom and Saudi Arabia [59]. Some exporters, such as Spain, show regional preferences, with a significant portion of their exports going to France, possibly due to regional trade agreements [59]. The interdependence between exporters and importers is clear, with the United Kingdom being a key destination for several exporters, including the United States, Egypt and Spain [60]. European countries, particularly Germany, the United Kingdom and France, are significant importers and rely on a mix of intra-European trade and imports from major producers such as the United States, Egypt and China [60]. This statistic shows the production value of sweet potatoes in the United States. According to a report, approximately 25.1 million cwt of potatoes were produced in the country in 2023 [60]. U.S. sweet potato exports, by fresh weight, increased by 1157% from 2001 to 2021, and the annual value of exports grew from UDS 14 million to USD 187 million over the same period [61]. According to the Food and Agriculture Organization of the United Nations (FAO), the United States was the largest global exporter, by volume, of sweet potatoes in 2020 [61]. Promotion of the health benefits of the root and the expansion of food companies’ sweet potato offerings, such as chips and French fries, helped drive the expansion [58]. Exports to the United Kingdom and the European Union experienced year-over-year growth from the mid-2000s through 2018 [59]. Over the past 20 years, major United States producing states have more than doubled sweet potato production to meet growing international and domestic demand.

The socioeconomic importance of sweet potatoes in Brazilian agriculture stems from their status as one of the most widely cultivated crops in the country, with approximately 24% of Brazilian municipalities producing sweet potatoes for both subsistence and commercial purposes [53]. The domestic market is primarily geared towards human consumption of fresh roots, though there is a growing market for processed products such as chips and glazed sweet potatoes [54]. Over the years, national productivity has shown a consistent upward trend. This increase in productivity is largely due to the adoption of recommended agricultural technologies, the use of high-yielding cultivars and the planting of healthy seedlings, all of which respond well to optimal production practices [55].

In recent years, sweet potatoes have become the fourth most consumed vegetable in Brazil. Although Brazil holds a modest position among the world’s largest exporters, Europe and Argentina remain its most reliable importers [60]. In 2019 alone, Brazil exported 8.8 thousand tons, as reported by the industry [46]. According to IBGE data, the sweet potato cultivation area in the state of São Paulo has tripled over the last decade, with production doubling, indicating an expansion trend in the coming years. While many countries prefer to import from nearby nations for fresher and higher-quality products, Brazilian sweet potatoes have gained a significant foothold internationally, with growth in export performance [53]. This success is mainly due to the country’s favorable climate, which supports year-round cultivation of sweet potato varieties, and the competitive pricing of Brazilian products, driven by the recent economic recession and currency devaluation [53]. The combination of lower prices, high quality and year-round availability has positioned Brazil as a strong competitor in the global sweet potato market [62].

This positive impact is due to using sweet potatoes in industrial biorefineries to produce high-value materials, including bioethanol, functional feed, antioxidants and food resources [62]. The non-profit Center for Science in the Public Interest (CSPI) has announced sweet potatoes as one of the ten “superfoods” for better health [54]. The United States Department of Agriculture (USDA) has also reported that sweet potatoes are the best bioenergy crop among starch crops on marginal lands, which does not affect food security. The Food and Agriculture Organization of the United Nations (FAO) estimated that the global population in 2050 will be 9.7 billion and require approximately 1.7 times more food than today. In this regard, sweet potatoes will be a solution to address issues such as food, energy, health and the environment [61]. The FAO has declared the sweet potato as one of the critical crops for global food security and the production of high-value-added biomaterials. Some events can directly and indirectly harm the growing sweet potato production, as many insect pests can potentially reduce the quality and yield of sweet potatoes.

## 7. Insect Pests

Insect pests are the primary sources of biotic stresses in crops [63]. These pests cause direct damage to plants by feeding on them and indirectly transmitting plant viruses, leading to large yield losses [64]. Global economic losses from insect pests in different crops are well documented. Hundreds of insects can cause severe damage to crops and are controlled by chemical pesticides, which are the primary sources of pollution and cause the development and progression of a series of health problems in humans and animals [65]. Because of these problems, scientists are always looking for alternative ways to control pests that harm sustainable agriculture. Estimated crop losses due to insect pests are around 32.1% globally, valued at over USD 470 billion [63]. Successful management of insect pests requires continuous application of harmful synthetic pesticides, which creates environmental pollution, health risks and resurgence of pests, causes the development of resistance, and increases the cost of cultivation [66]. There is a need to develop alternative and sustainable management practices to control insect pests and their losses [63]. The high productivity potential of sweet potatoes is restricted by a wide variety of insect pests that reduce the quantity and quality of their production. Among the pests that attack sweet potato crops in the USA and Brazil, five stand out as the most detrimental to production. These are highlighted in Table 1 for the USA and Table 2 for Brazil.

Beetles, *Cylas formicarius* (Fabricius, 1798), *Monocrepidius falli* (Lane, 1956) and *Diabrotica speciosa* (Germar, 1824) are significant insect pests of the sweet potato [2]. The damage caused by these insects is high [81]. The larvae feed on the pulp of sweet potato roots, causing holes and tunnels, and on the leaves [82]. This can result in root rot, making them unsuitable for consumption and sale [81]. Damaged areas often become entry points for pathogens, leading to secondary infections and deterioration of root quality [81]. Root damage can significantly reduce crop yield, affecting quantity and quality [81]. The attack can compromise plant growth, resulting in a reduction in plant vigor [82].

*Bemisia tabaci* (Gennadius, 1889), also known as white fly, is an insect pest affecting many crops, including sweet potatoes. The primary damage caused by *B. tabaci* includes sap-sucking, where it feeds on plant sap, which can cause general weakening, reduced growth and, in severe cases, death of plants [83]. In terms of virus transmission, the whitefly is a vector for several viruses, such as sweet potato mosaic virus and stem hardening virus [84]. These viruses can cause symptoms such as mosaic, mottling and leaf deformities, affecting plant health and productivity [85]. The accumulation of honeydew can promote the growth of sooty mold, a fungus that forms a black layer on the leaves, inhibiting photosynthesis and reducing plant vigor. Damage caused by sap-sucking and the presence of sooty mold can lead to a reduction in root quality and overall crop productivity [86].

*Spodoptera* is a genus of moths that includes several species of caterpillars, known to be significant pests in several crops, including sweet potatoes. The most common species of *Spodoptera* that affect sweet potatoes are *Spodoptera frugiperda* (Smith, 1797) (fall armyworm), *Spodoptera eridania* (Cramer, 1782) (southern armyworm) and *Spodoptera litura* (Fabricius, 1775) (tobacco caterpillar) [02]. The main damage caused by *Spodoptera* includes damage to leaves, resulting in defoliation and a reduction in the available photosynthetic area [02]. This can weaken the plants and reduce root production [87]. In addition to the leaves, these caterpillars can damage shoots and stems, compromising the growth and overall health of the plants [88]. Consumption of plant tissue can affect crop productivity. Pathogen transmission can create opportunities for secondary pathogens, exacerbating plant health problems [89].

**Table 2 insects-15-00823-t002:** The main pests that cause economic damage to sweet potato plantations in Brazil.

Insect Species	Order/Family	Common Name	Type of Damage	Distribution	Source
*Euscepes postfasciatus*	Coleoptera/Curculionidae	Sweet potato weevil	Roots/foliage/storage roots	South Pacific Islands, Japan, Taiwan, United States, Caribbean, Central, and South America.	[90,91,92,93]
*Megastes* spp.	Lepidoptera/Crambidae	Stem borer	Branches and roots	Costa Rica, Panama, Venezuela, Guyana, Suriname, Peru, Paraguay, Argentina, and Brazil.	[90,94,95,96,97,98]
*Bedellia somnulentella*	Lepidoptera/Bedelliidae	Sweet potato bold	Foliage	South America, North America, Europe, Asia, and Oceania.	[1,99,100,101,102,103,104,105]
*Spodoptera* spp.	Lepidoptera/Noctuidae	Fall armyworm	Foliage	South America, Central America, Africa, and Oceania.	[90,106]
*Diabrotica* spp.	Coleoptera/Chrysomelidae	Beetle	Foliage	South America, Central America, North America.	[90,107,108,109,110]

*Euscepes postfasciatus* (Fairmaire, 1849) and *Megastes* spp. are insect pests known as sweet potato borers. The larvae feed on sweet potato roots, causing tunnels and perforations [91]. This compromises the integrity of the roots, leading to rot and deterioration. Infestation can result in roots with poor appearance and compromised quality, affecting the commercial value and quantity of production [91]. Attack by the larvae can weaken the plants, making them more susceptible to secondary diseases and adverse conditions [92]. The genus *Diabrotica* includes several species of beetles known as “borers” or “cucurbit beetles,” which can cause significant damage to crops, including sweet potatoes [107]. The main species associated with damage to sweet potatoes is *Diabrotica speciosa* (Germar, 1824), known as the sweet potato borer [108]. The damage caused by these species is notable for its influence on the quality and productivity of the plants [80]. The decrease in the quality and quantity of roots can significantly reduce crop productivity [91]. *Bedellia somnulentella* (Zeller, 1847) is a moth of the Bedelliidae family, whose caterpillar is known to cause damage to sweet potatoes [100]. The larvae of *B. somnulentella* feed on the surface of the leaves, creating mines in young and mature leaves, initially small galleries followed by larger mines [100]. This damage can lead to a reduction in the photosynthetic area of the plants, affecting their growth and health [92]. Damage to the leaves can weaken the plants, making them more susceptible to secondary diseases and environmental stresses [91].

## 8. IPM

In Brazil, the sweet potato cultivation is considered relatively stable due to its low application of chemical inputs. However, this production’s environmental impact and carbon footprint can be significant, depending on the farming techniques employed, the use of fertilizers and pesticides, and the transportation and processing of the roots. Although this crop is naturally resilient, intensive irrigation, excessive pesticide use and monoculture can exacerbate its environmental impacts. Adopting sustainable practices, such as crop rotation, efficient irrigation systems, soil conservation and integrated pest management (IPM), is essential to mitigate these impacts. IPM is an effective and environmentally sensitive approach to pest management that relies on a combination of common-sense practices [111]. The widespread use of IPM results from concerns about the long-term viability of conventional agriculture [112].

IPM ensures sufficient, safe, equitable, stable flows of food and ecosystem services and greater agricultural profitability due to lower pest management expenses [112]. Yield losses caused by pests can be equivalent to the amount of food needed to feed almost 1 billion individuals when food security is considered [113]. Alternative means must be used to limit pest damage while avoiding the expense and adverse effects associated with synthetic pesticides [114]. However, excessive use of these pesticides brings additional obstacles, and it is now apparent that they should be avoided [114]. IPM involves the use of various pest control techniques designed to supplement or eliminate the use of synthetic pesticides [115]. It has been used for a long time and is a sustainable pest management method [115]. By applying IPM, pest populations are kept below a point where they cannot harm the economy [116]. It involves determining strategies that are practical, accessible, and minimize environmental damage [117]. IPM is a systematic strategy incorporating many pest control measures into one program. Incorporating cultural, biological, genetic, physical, legislative and mechanical restrictions decreases pesticide dependence [117].

The interpretation of IPM also varies among those who develop, promote, or practice IPM strategies. IPM programs use current and comprehensive information about pest life cycles and their ecological interaction [118]. This information, in combination with available pest control methods, is used to manage pest damage by the most cost-effective means and with the lowest possible risk to people, property and the environment [119]. When practicing IPM, producers know the potential for pest infestation and follow a four-tier approach [120]. The four steps include setting action limits, monitoring and identifying pests, and preventing and controlling them [120]. We focus on current IPM programs in high-value vegetable crops in the United States. IPM best practice in the US in plant systems includes routine monitoring of crops, use of mild chemicals (where the impact on benefit is known), and some monitoring of beneficial insects, and biological control is commonly used for greenhouse pests, but not to the same extent in the field [118]. Mechanical tools such as insect vacuums are used in high-value crops such as strawberries but are not an economical option in non-specialty crops and are not carbon efficient due to fossil fuel consumption [117]. IPM could be expanded and further integrated with cultural control practices to enhance its effectiveness [120]. New advancements and emerging trends in pest management provide opportunities to increase IPM adoption. Biological control agents play a critical role in IPM and contribute to plant protection by managing pest populations [120]. The biological control agents are generally specific for harmful organisms and do not kill beneficial microorganisms [120]. Table 3 and Table 4 below show the main biological control agents for the leading sweet potato pests.

Biological control is less costly and cheaper than any other method. Within IPM, one of the most modern approaches is the role of biological control in a system that must initially involve the biological balance of pests with their natural enemies and levels of economic damage [194]. The main advantages of these types of biological control are that they reduce negative impacts on human health, have minimal impact on the environment and reduce pest resistance [195]. With biological control, it is possible to suppress pest populations, making them less harmful. The natural enemies of insects are important in limiting potential pest densities [196]. These natural enemies include predators, parasitoids, pathogens or microbiological tools, namely biological control agents. The last are bacteria, fungi and others capable of providing beneficial results regarding pest damage and plant growth and health, respectively [197].

## 9. Conclusions

This article summarizes our current understanding of the following: sweet potato production in the United States and Brazil, as well as the primary states, the quantity of production in recent years, and exports to other countries; managing the main sweet potato pests, their geographic distribution, and their biological control agents; and the pests which feed on roots, with trade implications worldwide. For example, the sweet potato tuber shipments infested with the sweet potato weevil are generally not allowed for trade. There are several insect species which can attack at any stage in the growth and development of the crop. Indeed, the cultivation system and geographic locations greatly influence their prevalence. Controlling weevils with insecticides is generally difficult because they feed inside the roots. Using weevil-resistant sweet potato cultivars is a practical and economical method of control. The apparent inconsistency in resistance between cultivars from season to season or location to location complicates the development of weevil-resistant sweet potato varieties. Biological control reduces negative impacts on human health, has minimal effects on the environment and reduces pest resistance in sweet potato crops.

## Figures and Tables

**Figure 1 insects-15-00823-f001:**
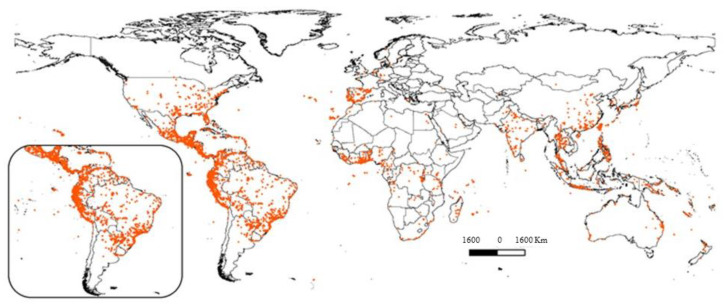
Regions in orange indicate where *Ipomoea batatas* (sweet potatoes) are planted in the world. Highlight is the primary center of origin of the species in Central and South America. Source: GBIF.

**Figure 2 insects-15-00823-f002:**
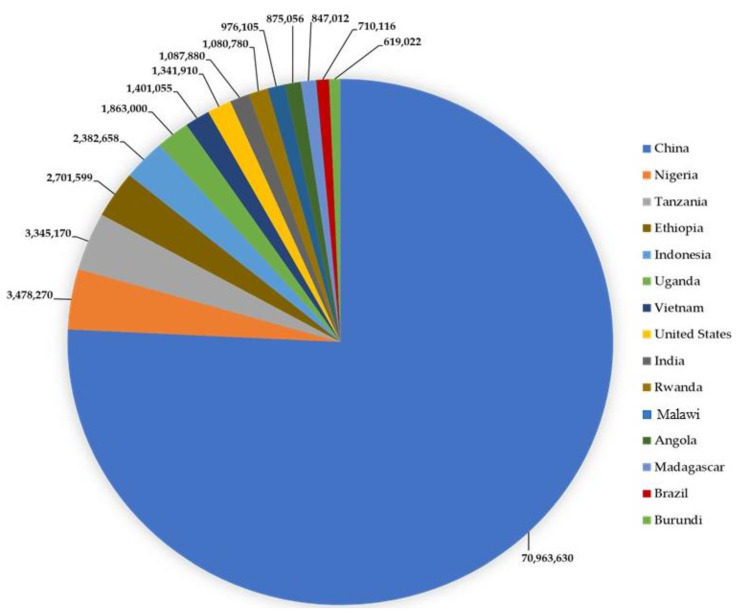
Annual global sweet potato production in metric tons [57,58].

**Figure 3 insects-15-00823-f003:**
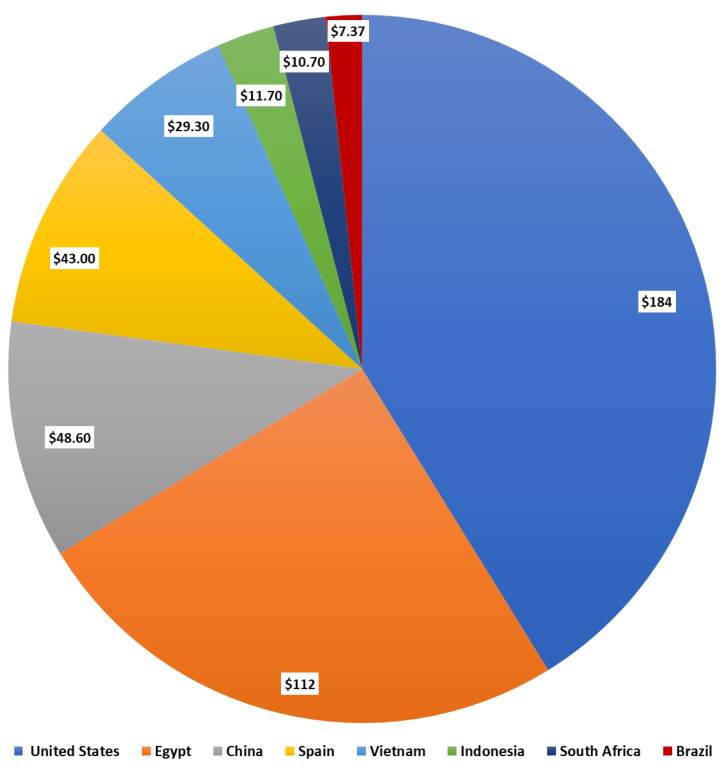
The economic impact of sweet potato exports on major countries in USD [59,60].

**Table 1 insects-15-00823-t001:** The main pests that cause economic damage to sweet potato plantations in the United States.

Insect Species	Order/Family	Common Name	Type of Damage	Distribution	Source
*Cylas formicarius*	Coleoptera/Brentidae	Sweet potato weevil	Roots/storage roots/foliage	Old world, southern USA, Greater Antilles, Central America, Hawaii.	[2,66,67]
*Monocrepidius falli*	Coleoptera/Elateridae	Southern potato wireworm	Roots/storage	South America, Central America, North America, Australia, New Zealand, Europe and Asia.	[2,68]
*Diabrotica speciosa*	Coleoptera/Chrysomelidae	Striped cucumber beetle	Roots/storage roots/foliage	Central, North, and South America, United States of America.	[2,69,70,71,72]
*Bemisia tabaci*	Hemiptera/Aleyrodidae	Sweet potato whitefly	Foliage	Global record, except in Antarctica.	[2,73,74,75,76,77]
*Spodoptera frugiperda*	Lepidoptera/Noctuidae	Fall armyworm	Foliage	Global record, except in Antarctica.	[2,78,79,80]

**Table 3 insects-15-00823-t003:** Biological control agents of the five significant pests of sweet potatoes in the United States.

Insect Species	Biological Control Agents	Source
*Cylas formicarius*	*Bracon mellitor* (Say)—(Hymenoptera: Braconidae)	[121,122,123,124,125,126,127,128,129,130]
	*Bracon punctatus* (Muesebeck)—(Hymenoptera: Braconidae)	
	*Metapelma spectabile* (Westwood)—(Hymenoptera: Metapelmatidae)	
	*Euderus purpureas* (Yoshimoto)—(Hymenoptera: Eulophidae)	
	*Rhaconotus* sp.—(Braconidae)	
	Ants—(Hymenoptera: Formicidae)	
	*Beauveria bassiana*—(Cordycipitaceae)	
	*Steinernema carpocapsae*—(Nematoda: Steinernematidae)	
	*Heterorhabditis bacteriophora*—(Nematoda: Heterorhabditidae)	
	*Metarhizium anisopliae*—(Hypocreales: Clavicipitaceae)	
*Monocrepidius falli*	Ground beetles—(Coleoptera: Carabidae)	[131,132,133,134,135,136,137,138]
	Rove beetles—(Coleoptera: Staphylinidae)	
	*Thereva nobilitata* (Fabricius) (Diptera: Therevidae)	
	*Enterococcus mundtii*—(Enterococcaceae)	
	*Staphylococcus pasteuri*—(Staphylococcaceae)	
	*Arthrobacter gandavensis*—(Micrococcaceae)	
	*Bacillus thuringiensis*—(Bacillaceae)	
	*Pseudomonas plecoglossicida*—(Pseudomonadaceae)	
	*Heterorhabditis bacteriophora*—(Heterorhabditidae)	
	*Macrocheles robustulus* (Mesostigmata: Macrochelidae)	
	*Gaeolaelaps aculeifer/Stratiolaelaps scimitus* (Mesostigmata: Laelapidae)	
	*Steinernema* sp.—(Steinernematidae)	
*Diabrotica speciosa*	*Metarhizium brunneum*—(Clavicipitaceae)	[139,140,141,142,143]
	*Beauveria bassiana*—(Cordycipitaceae)	
	*Pseudomonas protegens*—(Pseudomonadaceae)	
	*Pseudomonas chlororaphis*—(Pseudomonadaceae)	
	*Pseudomonas chlororaphis*—(Pseudomonadaceae)	
	*Celatoria diabrotica* (Shimer) (Diptera: Tachinidae)	
*Bemisia tabaci*	*Eretmocerus eremicus* Rose & Zolnerowich (Hymenoptera: Aphelinidae)	[144,145,146,147,148,149,150,151,152,153,154,155]
	Families—Aphelinidae/Azotidae/Encyrtidae	
	Signiphoridae (Chalcidoidea)	
	Platygastridae (Platygastroidea)	
	*Encarsia*/*Eretmocerus*—(Hymenoptera)	
	*Collops vittatus* (Say)—(Coleoptera: Melyridae),	
	*Nephaspis oculatus* (Blatchley)—(Coleoptera: Coccinellidae)	
	*Eretmocerus* sp.—(Aphelinidae)	
	*Delphastus catalinae* (Horn)—(Coleoptera: Coccinellidae)	
	*Hippodamia convergens* Guérin-Méneville (Coleoptera: Coccinellidae)	
	*Amblyseius swirskii* (Athias-Henriot) (Mesostigmata: Phytoseiidae)	
	Ladybugs	
	Lacewings	
	Phytoseiid Mites	
	Spiders	
	*Beauveria bassiana*—(Cordycipitaceae)	
	*Isaria fumosoroseus*—(Cordycipitaceae)	
	*Lecanicillium lecanii*—(Cordycipitaceae)	
	*Metarhizium anisopliae*—Clavicipitaceae	
	Laboulbeniales/Pyrenomycetes (Classe Ascomycota)	
	Hyphomycetes (Classe Deuteromycota)	
	Zygomycetes (Classe Zygomycota)	
	Hyphomycetes (Classe Deuteromycota)	
	*Verticillium/Isaria/Aschersonia*	
*Spodoptera frugiperda*	*Cotesia marginiventris* (Cresson)—(Hymenoptera: Braconidae)	
	*Chelonus insularis* (Cresson)—(Hymenoptera: Braconidae)	
	*Campoletis grioti* (Blanchard)—(Hymenoptera: Ichneumonidae)	
	*Meteorus autographae* Muesebeck—(Hymenoptera: Ichneumonidae)	
	*Coenochilus bifoveolatus* Schein—(Coleoptera: Scarabaeidae)	
	*Taquinídea incerta* (Linnaeus, 1758) (Diptera: Tachinidae)	
	*Ophion* spp.—(Hymenoptera: Ichneumonidae)	
	*Haematochares obscuripennis* (Stal)—(Heteroptera: Reduviidae)	
	*Peprius nodulipes* (Signoret) (Heteroptera: Reduviidae)	
	*Pheidole megacephala* (F.) (Hymenoptera: Formicidae)	
	*Coccygidium Luteum* (Brullé) (Hymenoptera: Braconidae)	
	*Trichogramma* spp. (Hymenoptera: Trichogrammatidae)	
	*Charops* spp. (Hymenoptera: Ichneumonidae)	
	*Drino quadrizonula* Thomson—(Diptera: Tachinidae)	
	*Telenomus remus* Nixon—(Hymenoptera: Scelionidae)	
	*Coccygidium luteum* (Brullé)—(Hymenoptera: Braconidae)	[156,157,158,159,160,161,162,163,164,165,166,167,168,169,170,171,172,173,174]
	*Bacillus thuringiensis*	
	*Metarhizium anisopliae*	
	*Beauveria bassiana*	
	*Baculovirus*	
	*Nomuraea rileyi*	
	*Steinernema feltiae*	
	*Steinernema carpocapsae*	
	*Heterorhabditis indica*	

**Table 4 insects-15-00823-t004:** Biological control agents of the five major sweet potato pests in Brazil.

Insect Species	Biological Control Agents	Source
*Eucepes postfasciatus*	*Catolaccus grandis* (Burks) (Hymenoptera: Pteromalidae)	[175,176,177,178,179]
	*Bracon yasudai* Maeto e Uesato (Hymenoptera: Braconidae)	
*Megastes* spp.	Family Chalcididae	[180,181,182]
	Family Braconidae	
*Bedellia somnulentella*	*Conura* sp. (Hymenoptera: Chalcididae)	[102,183,184,185]
	*Horismenus cupreus* (Ashmead, 1894) (Hymenoptera: Eulophidae)	
	*Protonectarina sylveirae* (Saussure, 1854) (Hymenoptera: Vespidae)	
	*Agelaia vicina* (Saussure, 1854) (Hymenoptera: Vespidae)	
*Diabrotica* spp.	*Beaveria bassiana*	[186,187]
	*Metharhizium anisopliae*	
	Entomopathogenic nematodes	
*Spodoptera* sp.	*Podisus nigrispinus* (Dallas) (Hemiptera: Pentatomidae)	[188,189,190,191,192,193]
	*Telenomus* spp. (Hymenoptera: Platygastridae)	
	*Trichogramma* spp. (Hymenoptera: Trichogrammatidae)	
	*Campoletis sonorensis* (Cameron) (Hymenoptera: Ichneumonidae)	
	*Chelonus* sp. (Hymenoptera: Braconidae)	
	*Trichospilus diatraeae* Cherian & Margabandhu (Hymenoptera: Eulophidae)	
	*Doru luteipes* (Scudder) (Dermaptera: Forficulidae)	
	*Beauveria bassiana* (Bals.)	
	*Metharizium anisopliae* (Metsch.)	
	Entomopathogenic nematodes	

## Data Availability

The data are available from the corresponding author (M.H.) upon request.

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
