# Peer review of "Major Insect Pests of Sweet Potatoes in Brazil and the United States, with Information on Crop Production and Regulatory Pest Management"

_insects, 2024, doi:10.3390/insects15100823_

Round 1

Reviewer 1 Report

Comments and Suggestions for Authors

This paper provides information on the production and Major insect pests of sweet potatoes in the US and Brazil. After revision, the manuscript can be published in the Insects Journal. Here are my suggestions:

The title does not entirely match the context of the paper. The manuscript mainly focuses on the production of sweet potatoes in the US and Brazil. I recommend changing the paper title to a more inclusive title, which includes both production and insect pests, which is what is reflected in the paper.  

The IPM section has general information on IPM. Please search for any papers published on IPM in sweet potato production in the US and Brazil (if any) and add them to the text.

Line 14: quratintine should be changed to quarantine.

Line 86: I suggest writing the headline: Sweet Potato (Ipomoea batatas (L.) Lam). Please recheck the author’s name.

Line 88: the references should be merged.

Line 172: the purpose of this sentence is unclear: In 2021, it produced over 4.3 million pounds (100 pounds), according to the Mississippi Department of Agriculture and Commerce. What does the (100 pounds) mean here?

Line 176: Have you explained cwt in the text before? If not, please provide information on what these words stand for.

Line 376: Please provide the complete name of the genus for C. formicarius and the author’s name for all species when mentioning them in the whole text for the first time.

Line 383: Please remove white scale and use white fly.

Tables 1 and 2: please remove the “are and period: at the end of the caption.

Line 420: remove period before reference.

Tables 3 and 4: remove the period from the end

Line 450: IPM best practice.

Author Response

Comments 1: The title does not entirely match the context of the paper. The manuscript mainly focuses on the production of sweet potatoes in the US and Brazil. I recommend changing the paper title to a more inclusive title, which includes both production and insect pests, which is what is reflected in the paper.  

Response 1: Thank you for pointing this. We agree, was made: Major Insect Pests of Sweet Potatoes in Brazil and the United States, with Information on Regulatory Pest Management and Crop Production

Comments 2: The IPM section has general information on IPM. Please search for any papers published on IPM in sweet potato production in the US and Brazil (if any) and add them to the text.

Response 2:

Comments 3: Line 14: quratintine should be changed to quarantine.

Response 3: As advised the spellings are corrected to quarantine.

Comments 4: Line 86: I suggest writing the headline: Sweet Potato (Ipomoea batatas (L.) Lam). Please recheck the author’s name.

Response 4: Modified to Sweet potato (Ipomoea batatas (L.) Lam).

Comments 5: Line 88: the references should be merged.

Response 5: The references have been merged as [1,17,18].

Comments 6: Line 172: the purpose of this sentence is unclear: In 2021, it produced over 4.3 million pounds (100 pounds), according to the Mississippi Department of Agriculture and Commerce. What does the (100 pounds) mean here?

Response 6: A suggested, these units have been modified to metric tons.

Comments 7: Line 176: Have you explained cwt in the text before? If not, please provide information on what these words stand for.

Response 7: Thank you for pointing this. A hundredweight (abbreviated as CWT) is a standard unit of weight or mass used being used. 

Comments 8: Line 376: Please provide the complete name of the genus for C. formicarius and the author’s name for all species when mentioning them in the whole text for the first time.

Response 8: Thank you for pointing this. All scientific names with authors' have been inserted in the text.

Comments 9: Line 383: Please remove white scale and use white fly.

Response 9: As suggested, this has been modified to white fly.

Comments 10: Tables 1 and 2: please remove the “are and period: at the end of the caption.

Response 10: This has been removed.

Comments 11: Line 420: remove period before reference.

Response 11: This has been removed.

Comments 12: Tables 3 and 4: remove the period from the end

Response 12: This has been removed.

Comments 13: Line 450: IPM best practice.

Response 13: As advised, it has been inserted.

Reviewer 2 Report

Comments and Suggestions for Authors

The topic of the paper review is interesting, as it provides a comprehensive overview of sweet potato production in the United States and Brazil, two of the leading producers of this important food crop. This review also examines the economic impact of sweet potato production and trade and the challenges posed by insect pests. The paper is well-written and easy to read. The conclusions are consistent with the evidence and arguments presented. The paper also includes several tables and figures that add to the paper and aid understanding.

Here are some possible areas of improvement for the paper:

The study is well-structured and the data visualizations (tables and figures) are well-presented The authors suggested more research into biological pest control methods that align with the increase in organic agriculture.   However, the manuscript could deeply describe the environmental impact of sweet potato production, including water usage, land use changes, and the carbon footprint associated with farming practices. This would improve the discussion
Additionally, while the study focuses on the U.S. and Brazil, briefly mentioning other major producers and their pest management strategies could provide a broader context.
Overall, the study is informative and relevant, contributing to the understanding of sweet potato production and pest management in two key agricultural regions.

Author Response

Comments 1: The study is well-structured and the data visualizations (tables and figures) are well-presented. The authors suggested more research into biological pest control methods that align with the increase in organic agriculture.   However, the manuscript could deeply describe the environmental impact of sweet potato production, including water usage, land use changes, and the carbon footprint associated with farming practices. This would improve the discussion
Additionally, while the study focuses on the U.S. and Brazil, briefly mentioning other major producers and their pest management strategies could provide a broader context.
Overall, the study is informative and relevant, contributing to the understanding of sweet potato production and pest management in two key agricultural regions.

Response 1: Thank you for the excellent comments and suggestions. As advised, additional information has been inserted regarding the environmental impacts of sweet potato cultivation and its carbon footprint has been included in item 8, lines 441-448. The major sweet potato producers are mentioned in item 5, Global sweet potato production, and Figure 1 presents the distribution and planted area worldwide. These sections provide a comprehensive overview of global production and the environmental aspects associated with the cultivation of this crop, highlighting both its economic importance and the sustainability challenges it faces.

Reviewer 3 Report

Comments and Suggestions for Authors

Manuscript ID: insects-3245083 entitled “Major Insect Pests of Sweet Potatoes in Brazil and the United States, with Information on Regulatory Pest Management” by Maria J. Santos Cabral, Muhammad Haseeb, Marcus Alvarenga Soares submitted to Insect Pest and Vector Management. I found the review is well written on the sweet potatoes production and importance parts in the United States and Brazil. However, the insect pest information part is not reviewed in depth. With many mistakes in the tables that should be corrected before consideration for publication. I added my notes to the attached PDF and some of my concerns are:

·         Insect pest species in the tables are not specified correctly according to recent nomenclature. The main pests that cause economic damage to sweet potato plantations in the United States (Striped cucumber beetle) should match Acalymma vittatum not the synonym Diabrotica sp., Fall armyworm should match Spodoptera frugiperda not Spodoptera sp., Southern potato wireworm should match Monocrepidius falli (Lane) not Conoderus sp.

·         Scientific names should be fully reported with author, order, and family when first mentioned in text.

·         Table 2 insects should be specified, what is Cow beetle has do in this table.

·         The primary center of origin of the species is not well defined.

·         Change Walawi to Malawi in Figure2, add what the $ values means in Figure 3.

·         Incorrect information in table 3: Metapelma spectabile (Westwood) is Family Metapelmatidae not Braconiade.

·         Table 3. Biological control agents of the five significant pests of sweet potatoes in the United States. In the table there is 22 names of plant species with no explanation on what these species have to do with biological control agents of sweet potatoes in the United States.

·         Bedbug is listed as biocontrol agent of Bemisia tabaci.

·         Cow beetle used incorrectly as a common name for Diabrotica (Table 2).

·         Carabídeos beetles used as a common name for the ground beetles (Table 3).

Author Response

Comments 1: Insect pest species in the tables are not specified correctly according to recent nomenclature. The main pests that cause economic damage to sweet potato plantations in the United States (Striped cucumber beetle) should match Acalymma vittatum not the synonym Diabrotica sp., Fall armyworm should match Spodoptera frugiperda not Spodoptera sp., Southern potato wireworm should match Monocrepidius falli (Lane) not Conoderus sp.

Response 1: Thanks for the comments & suggestions. As advised, the nomenclatures have been revised.

Comments 2:  Scientific names should be fully reported with author, order, and family when first mentioned in text.

Response 2: The authors and scientific names have been properly reported.

Comments 3:    Table 2 insects should be specified, what is Cow beetle has do in this table.

Response 3: Thanks for the comment. The cow beetle is a common pest of sweet potatoes in Brazil. Correction has been made.

Comments 4: The primary center of origin of the species is not well defined.

Response 4: Thanks for the comment. Correction has been made to South America.

Comments 5: Change Walawi to Malawi in Figure2, add what the $ values means in Figure 3.

Response 5: These changes have been made.

Comments 6: Incorrect information in table 3: Metapelma spectabile (Westwood) is Family Metapelmatidae not Braconiade.

Response 6: This has been corrected.

Comments 7: Table 3. Biological control agents of the five significant pests of sweet potatoes in the United States. In the table there is 22 names of plant species with no explanation on what these species have to do with biological control agents of sweet potatoes in the United States.

Response 7: As suggested, the irrelevant information of the species has been removed.

Comments 8: Bedbug is listed as biocontrol agent of Bemisia tabaci.

Response 8: This has been resolved.

Comments 9:  Cow beetle used incorrectly as a common name for Diabrotica (Table 2).

Response 9: This has been corrected.

Comments 10: Carabídeos beetles used as a common name for the ground beetles (Table 3).

Response 10: As advised, this has been corrected.

Round 2

Reviewer 3 Report

Comments and Suggestions for Authors

Authors revised the manuscript according to reviewers' suggestions. 

The manuscript is now readable and ready for consideration by editor as presented, I added some minor notes to the attached PDF for authors' final revision.  

Author Response

Dear Reviewer: 

Thank you for your time and review of our manuscript. We greatly appreciate this. As advised, the following revisions have been made in the attached manuscript file:

Line 168: deleted ‘is’

Reference 2: Applied bold to the year.

Reference 13: Acta Horticulturae 583: 143 – 154 added.

Reference 28 & 29: Correct year (2024) was inserted.

Reference 57: Applied bold to the year.

Reference 60: Report number: 194 inserted.

Reference 86: Page numbers 93 – 243 inserted.

Reference 92: Insert the additional information for the reference as: IPN: EENY129. Gainesville: Institute Food and Agricultural Sciences. from https://edis.ifas.ufl.edu/in286.

References:  111, 121, 134, and 141: Corrected with a long hyphen.

Reference 155: Year inserted.

Reference 203: 29 August 2024 inserted.
